# ARE VISION FOUNDATION MODELS FOUNDATIONAL FOR ELECTRON MICROSCOPY IMAGE SEGMENTATION?

**Caterina Fuster-Barceló**
Department of Molecular Life Sciences,
Universität Zürich
Zürich, Switzerland
caterina.fusterbarcelo@mls.uzh.ch

**Virginie Uhlmann**
Department of Molecular Life Sciences,
Universität Zürich
Zürich, Switzerland
virginie.uhlmann@mls.uzh.ch

## ABSTRACT

Although vision foundation models (VFMs) are increasingly reused for biomedical image analysis, it remains unclear whether the latent representations they provide are general enough to support effective transfer and reuse across heterogeneous microscopy image datasets. Here, we study this question for the problem of mitochondria segmentation in electron microscopy (EM) images, using two popular public EM datasets (Lucchi++ and VNC) and three recent representative VFMs (DINOv2, DINOv3, and OpenCLIP). We evaluate two practical model adaptation regimes: a frozen-backbone setting in which only a lightweight segmentation head is trained on top of the VFM, and parameter-efficient fine-tuning (PEFT) via Low-Rank Adaptation (LoRA) in which the VFM is fine-tuned in a targeted manner to a specific dataset. Across all backbones, we observe that training on a single EM dataset yields good segmentation performance (quantified as foreground Intersection-over-Union), and that LoRA consistently improves in-domain performance. In contrast, training on multiple EM datasets leads to severe performance degradation for all models considered, with only marginal gains from PEFT. Exploration of the latent representation space through various techniques (PCA, Fréchet Dinov2 distance, and linear probes) reveals a pronounced and persistent domain mismatch between the two considered EM datasets in spite of their visual similarity, which is consistent with the observed failure of paired training. These results suggest that, while VFMs can deliver competitive results for EM segmentation within a single domain under lightweight adaptation, current PEFT strategies are insufficient to obtain a single robust model across heterogeneous EM datasets without additional domain-alignment mechanisms.

## 1 INTRODUCTION

Over the past decade, deep learning (DL) has become a central tool for quantitative microscopy image analysis, offering outstanding performance across tasks as diverse as image reconstruction, object segmentation, feature detection, and classification (Moen et al., 2019; Cunha et al., 2024). At the same time, the ecosystem of publicly-available generalist models pre-trained on large datasets of diverse microscopy images has expanded rapidly, ranging from task-specific architectures trained for specific problems (Schmidt et al., 2018; Stringer et al., 2021; Weigert et al., 2018) to large-scale vision and vision-language foundation models (FMs) intended to be general enough for broad reuse (Zhao et al., 2025; Archit et al., 2025). These developments have shifted the bottleneck for the adoption of DL approaches in microscopy image analysis: instead of designing new problem- and data-specific architectures, the practical hurdle now resides in determining *when* and *how* FMs can be safely used and, if required, adapted to new microscopy image data and biological questions.

A major challenge comes from the fact that microscopy data exhibit substantial distribution shift, even within the same imaging modality (Stacke et al., 2020). Differences in sample preparation, acquisition protocols, resolution, contrast, and any other factor affecting the imaging itself can result in subtle dataset-specific signatures that hamper the performance generalization of DL models (Abdul-

lah et al., 2014). As a result, it is hard to ensure that models are able to learn meaningful representations that remain stable across potentially big yet visually imperceptible domain shifts. This issue is also encountered in other, non-image data modalities, as illustrated by recent work questioning how "foundational" FMs truly are for time-series forecasting, and demonstrating that generalization can collapse when the statistics of the original training and target domains are misaligned (Karaouli et al., 2025). Motivated by this perspective, we here investigate an analogous question for vision foundation models (VFMs) used to analyse electron microscopy (EM) images, and explore whether current VFMs are foundational enough to support transfer across heterogeneous EM datasets or if they instead remain strongly domain-aware. More specifically, we study transfer for the task of mitochondria segmentation in EM relying on two widely-used public datasets, Lucchi++ (Lucchi et al., 2011; Casser et al., 2020) and VNC (Gerhard et al., 2013), and considering three recent representative VFMs, DINOv2 (Oquab et al., 2023), DINOv3 (Siméoni et al., 2025), and OpenCLIP (Cherti et al., 2023). Rather than evaluating full end-to-end fine-tuning pipelines, we focus on two practical reuse settings that preserve most of the pre-trained weights: a frozen-backbone regime with a lightweight segmentation head, and parameter-efficient fine-tuning (PEFT) via low-rank adaptation (LoRA, Hu et al. (2022)).

We organize our analysis around three questions: (1) Can VFMs generalize beyond their pre-training distribution in EM image data, (2) can PEFT improve generalization across EM datasets, and (3) are VFMs with and without PEFT competitive with task-specific models and other end-to-end alternatives in practice. Across all backbones, we find that frozen-backbone adaptation performs well when training and testing on a single EM dataset, and that LoRA further improves in-domain performance. However, when training on the union of the two considered EM datasets, performance collapses for all the models we studied, and PEFT only yields marginal gains. Further representation-space diagnostics (principal component analysis (PCA), Fréchet distance computed in DINOv2 feature space (Stein et al., 2023), and linear probes) indicates that Lucchi++ and VNC remain strongly separable even after PEFT, suggesting that inter-dataset domain shift, despite the two being composed of EM images, is a dominant factor in underperformance.

**Contributions.** Our main contributions in this work are as follows: **(1)** we provide a controlled benchmark of three major VFMs on an EM segmentation task under frozen-backbone and LoRA-PEFT adaptation; **(2)** we provide empirical evidence that paired training across heterogeneous datasets from the same microscopy imaging modality severely fails; **(3)** we carry out domain-mismatch analyses to link this failure to persistent representation separation across individual datasets; **(4)** we report a performance comparison against representative non-VFM baselines from the EM segmentation literature. The code developed for this work is publicly available at `https://github.com/uhlmanngroup/MicroscopyFoundationHub`.

## 2 RELATED WORK

Several works have recently started to explore whether FM paradigms can be meaningfully instantiated for specific biomedical imaging domains. For example, a recent multimodal model proposed for digital pathology has been explicitly positioning itself as an FM by integrating heterogeneous data modalities under a unified representation (Ding et al., 2025). In contrast, truly EM-native FMs remain relatively scarce. While a recent work frames itself as an FM approach for EM (Yu et al., 2025), it only provides brief downstream evaluations. Furthermore, the reported performance of this model for mitochondria segmentation appears to be surprisingly low (mean IoU$< 60\%$), suggesting that robust transfer for segmentation in EM remains an open challenge. In practice, the majority of the recent EM segmentation literature either still follows a paradigm in which task-specific models are trained end-to-end for a given dataset (Franco-Barranco et al., 2022), or seeks to adapt general-purpose VFMs through fine-tuning and related strategies (Archit et al., 2025; Zhou et al., 2024).

Recent efforts have investigated how pre-trained representations can be leveraged for EM segmentation, particularly for mitochondria. Two models have been proposed by He et al. (2025) specifically for mitochondria segmentation, namely EM-DINO, a DINO-style encoder pre-trained on EM data, and OmniEM, a unified dense-prediction model that builds on the EM-DINO embeddings. OmniEM reports improved mitochondria segmentation across diverse EM benchmarks, including the Lucchi++ dataset considered here. In parallel, DINOSim (González-Marfil et al., 2025) exploits DINOv2 patch embeddings and lightweight, non-parametric inference (*e.g.*, similarity- and $k$NN-

based labelling) to perform zero-shot semantic segmentation on EM datasets, including Lucchi++ and VNC. The reported results highlight that fully zero-shot pipelines still offer a substantially poorer performance than supervised baselines, motivating the systematic study of when and why VFMs successfully transfer to EM. Beyond DINO-style encoders, other methods for self-supervised pre-training with GAN objectives on unlabelled EM data have been proposed and demonstrated consistent gains for downstream tasks, including semantic segmentation (Kazimi et al., 2024). Finally, SAM-centric domain adaptation approaches such as SAMDA (Wang & Xiao, 2025) combine the Segment Anything Model (SAM) (Ravi et al., 2024) with a U-Net (Ronneberger et al., 2015) or nnU-Net-style (Isensee et al., 2018) expert component to improve transfer under limited supervision, further illustrating that domain shift remains a bottleneck even within a single microscopy imaging modality such as EM.

A particularly active line of work focuses on adapting SAM to biomedical image data (Archit et al., 2025), motivated by the observation that SAM's off-the-shelf performance often degrades under microscopy-specific distributions. Recent studies have explored parameter-efficient adaptation of SAM, including LoRA-based tuning, for object segmentation in biomedical images (Teuber et al., 2025). Other approaches, such as mixtures of PEFT experts with gating mechanisms, have also been proposed to specialize SAM to different domains and tasks (Sahay & Savakis, 2025). In parallel, there is a growing trend towards the reuse of general-purpose VFMs trained on natural images as feature encoders for biomedical data. OpenCLIP embeddings have been used as a representative feature space to condition diffusion models for generating microscopy images (Siemens et al., 2023), and CLIP-style encoders have been adapted for downstream tasks such as multilabel classification in histopathology image data (Bai & Miyata, 2025). Similarly, DINO-family encoders are increasingly used in microscopy pipelines, as exemplified by DINOv2-based self-supervised models for 3D microscopy segmentation and tracking (Lavaee et al., 2026), as well as by recent benchmarks evaluating DINOv3 as a transferable backbone across medical vision tasks (Liu et al., 2025). Given this growing body of work on SAM adaptation and domain-specific refinement, we chose not to centre our evaluation on SAM and instead focus on the comparatively less studied but widely deployed VFMs models DINOv2/DINOv3 and OpenCLIP to assess whether general-purpose pre-training on natural images can support robust EM segmentation and, crucially, whether lightweight adaptation suffices to bridge inter-dataset shifts within EM imaging.

## 3 METHODOLOGY

### 3.1 DATASETS

We use two EM datasets, VNC (Gerhard et al., 2013) and Lucchi++ (Lucchi et al., 2011; Casser et al., 2020), selected for their wide adoption in the EM segmentation literature, which enables a direct comparison against a broad range of established methods for mitochondria segmentation in EM image data. While both datasets focus on mitochondria segmentation in EM, they represent distinct imaging domains: VNC consists of serial-section TEM images from the Drosophila ventral nerve cord, whereas Lucchi++ uses FIB-SEM volumes from the mouse hippocampus. Despite these differences in instrument, specimen, and acquisition settings, which all introduce low-level domain mismatches, using them in tandem allows us to investigate the challenge of building a generalist model for mitochondria segmentation in EM images. We provide in Supplementary Section A.1 a detailed description of the content and format of these two datasets.

### 3.2 MODELS

#### 3.2.1 VISION FOUNDATION MODEL BACKBONES

We selected the 3 widely used VFMs DINOv2 (Oquab et al., 2023), DINOv3 (Siméoni et al., 2025), and OpenCLIP (Cherti et al., 2023), as they span complementary pre-training objectives (self-supervised vision pre-training and vision-language pre-training) and offer publicly available weights as well as robust, documented, and reproducible implementations. We provide an in-depth rationale for this selection in Supplementary Section A.2. For all backbones, we primarily report results on the large version (L) of the underlying ViT model, although we observed that scaling it up or down does not have a major impact on results (Supplementary Section A.3).

### 3.2.2 SEGMENTATION HEAD ARCHITECTURE

The considered VFM backbones tokenize an input image into non-overlapping patches and process a sequence of patch tokens, which we reshape into a 2D feature map of size $B \times D \times H' \times W'$, where $B$ is the batch size, $D$ is the backbone embedding dimension, and $(H', W')$ is the spatial patch grid, which depends on the (pre-processed) input resolution and on the model patch size $P \in \{14, 16\}$. The embedding dimension $D$ is not tuned independently, but is instead fixed by the chosen backbone variant (ViT-S/B/L/...) and corresponds to the token width of the vision encoder (pre-projection for OpenCLIP). For example, for the ViT-L variants used in our experiments, $D = 1024$ for DINOv2/DINOv3, whereas OpenCLIP ViT-L/14 uses $D = 768$ (with smaller variants typically using $D = 384$ for ViT-S and $D = 768$ for ViT-B). We then attach a segmentation head composed of a lightweight convolutional decoder on top of these embeddings to obtain dense pixel-wise predictions. This design keeps the segmentation head largely backbone-agnostic, as it only requires access to the patch-token grid and its channel dimension.

Importantly, we use the same segmentation head architecture and optimization protocol across all backbones and adaptation regimes. The only backbone-specific change is the input-channel size to match the backbone embedding dimension $D$ and patch grid. In the frozen-backbone setting, only the segmentation head parameters are optimized while the backbone is kept fixed. Under PEFT, the segmentation head remains unchanged, and we additionally optimize the injected adapter parameters in the backbone. This ensures that performance differences across methods can be fully attributed to the backbone representations and adaptation strategy, rather than to changes in the model.

## 3.3 TRAINING AND ADAPTATION PROTOCOLS

We adopt a standardized training protocol across datasets and backbones to ensure that the observed differences in performance arise from the representation and adaptation regime rather than from optimization details. Unless stated otherwise, we optimize a Dice loss relying on the MONAI library implementation (Cardoso et al., 2022) and train for up to 1000 epochs with early stopping (patience$= 20$), triggered by the validation loss. For each run, we select the checkpoint achieving the lowest validation loss and report test performance from that checkpoint. Validation splits are created by randomly holding out $10\%$ of the available training data. We use the AdamW (Loshchilov & Hutter, 2017) optimizer with learning rate $5 \times 10^{-5}$ and weight decay $10^{-4}$, and, following standard practice, apply no data augmentation as FMs are expected to have already learned robust features from their large-scale pre-training with extensive augmentation (Oquab et al., 2023). Due to the memory footprint of the backbones, we use a batch size of 2 for all experiments. The number of iterations per epoch therefore depends on the size of the dataset, with $\lceil N_{\text{train}}/2 \rceil$ update steps per epoch. We report results as the mean and standard deviation (std) averaged across multiple runs. We normalize the image inputs using ImageNet statistics (mean and std) (Mishkin et al., 2017) for both training and validation, and do not apply any dataset-specific intensity normalization as expected for VFMs used in this setting. All experiments were run on our institutional high-performance compute cluster using a single NVIDIA H100 GPU per run.

### 3.3.1 FROZEN BACKBONE (HEAD-ONLY) ADAPTATION

As a baseline, we adapt each VFM without updating its pre-trained backbone weights. Specifically, we keep the backbone frozen (including normalization layers), and only optimize the segmentation head parameters. Although this setting is not strictly "zero-shot" (since it learns a task-specific decoder), it provides a low-capacity and comparable adaptation across backbones: the same segmentation head architecture, loss, optimizer, and training schedule are used, and the backbone representation is held fixed. This design allows us to isolate the contribution of the pre-trained representation quality to the final segmentation performance.

### 3.3.2 PEFT WITH LoRA

To test whether limited backbone adaptation improves the robustness of performance under domain shift, we additionally employ LoRA (Hu et al., 2022) as PEFT strategy. We provide an in-depth explanation of how LoRA is injected into the backbones in Supplementary Section A.4. In this regime, the segmentation head is trained exactly as in the frozen-backbone baseline, while the trainable parameter set is augmented by the LoRA adapters. While carrying out LoRA increases the

total number of trainable parameters as compared to head-only adaptation, it remains relatively lightweight, especially considering the size of the VFM backbones (Supplementary Section A.5).

### 3.3.3 SINGLE DATASET VS. PAIRED DATASET TRAINING REGIMES

We evaluate transfer under two training regimes: "single dataset" training, where models are trained on Lucchi++ or VNC individually using their corresponding splits, and "paired dataset" training, where models are trained on the union of both datasets to assess whether a single backbone can support generalized mitochondria segmentation in EM image data. For paired training, we intentionally use the natural (unbalanced) dataset mixture without reweighting or resampling to evaluate transfer under realistic data availability. We always report performance on the test datasets, both in the single and paired dataset training scenarios.

A practical complication in the paired dataset training scenario is that the datasets have different native image dimensions and the backbones use different patch sizes. Instead of cropping the input data, we rescale each image to a backbone-specific resolution by preserving its aspect ratio and setting the longest edge to a fixed target, then snapping both sides to a multiple of the patch size. This "longest-edge with patch-multiple" resizing strategy is applied only in the paired regime to avoid padding artifacts and yield inputs that are compatible with patch tokenization while preserving as much of the field of view as possible.

## 3.4 DOMAIN MISMATCH ANALYSIS

To support the hypothesis that training on paired datasets fails due to persistent domain mismatch between Lucchi++ and VNC, we performed two lightweight diagnostics in the representation space provided by the backbone VFM. For each image, we extract a fixed-dimensional embedding from the VFM (either obtained with frozen or PEFT-adapted weights) by pooling the final-layer patch tokens into a single feature vector. In our implementation, we used a simple global pooling operation on the patch token grid, but observed that other standard choices (*e.g.*, using the CLS token instead) lead to the same qualitative conclusions.

**PCA visualization and Fréchet DINOv2 distance.**  We first visualize the joint embedding space by applying PCA to the pooled features and projecting both datasets onto the first two leading principal components. To quantify distributional mismatch, we compute a Fréchet distance between the two feature distributions in the same spirit as the Fréchet Inception Distance (Heusel et al., 2017), but using DINOv2 features instead of Inception features, hence referred to as Fréchet DINOv2 Distance (FD-DINOv2, Stein et al. (2023), Supplementary Section A.6). We then report FD-DINOv2 for the frozen and LoRA-adapted backbones to assess whether PEFT reduces the representation gap.

**Linear separability of domains.**  As a complementary readout, we also train a logistic regression (LR) classifier to predict the dataset label (Lucchi++ vs. VNC) from the extracted embeddings, using the same train/validation split strategy as in the segmentation experiments. We then compare the performance of this linear probe for features extracted with the frozen and LoRA-adapted backbones to determine whether PEFT meaningfully decreases domain discriminability.

## 3.5 PERFORMANCE EVALUATION METRICS

We computed performance metrics over the full evaluation split in a single pass using *dataset-level* pixel totals rather than averaging per-image scores to avoid the high variance that could arise when mitochondria occupy very different fractions of the overall image across slices. We evaluate segmentation performance using the foreground IoU ($IoU_{fg}$, Supplementary Section A.6), where all non-zero labels are collapsed into a single foreground class. The foreground IoU is a standard metric in the EM segmentation literature to avoid biasing the metric towards the background class, which is easier to predict in images where the foreground is sparse (de Andrade & Boccato, 2024). It also enables direct comparison to prior work on EM segmentation (González-Marfil et al., 2025).

| Backbone | Lucchi++ | VNC | Paired |
|---|---|---|---|
| DINOv2-L/14 | $\mathbf{0.717 \pm 0.006}$ | $0.723 \pm 0.002$ | $0.053 \pm 0.005$ |
| DINOv3-L/16 | $\mathbf{0.717 \pm 0.000}$ | $\mathbf{0.833 \pm 0.000}$ | $\mathbf{0.054 \pm 0.000}$ |
| OpenCLIP ViT-L/14 | $0.568 \pm 0.003$ | $0.573 \pm 0.011$ | $0.032 \pm 0.003$ |

Table 1: **Frozen-backbone (head-only) segmentation performance.** Foreground IoU ($\text{IoU}_{\text{fg}}$) on Lucchi++ and VNC under single-dataset and paired training, reported and the mean±std over 5 runs. In the paired setting, the reported $\text{IoU}_{\text{fg}}$ is a macro-average across both datasets.

## 4 RESULTS

### 4.1 GENERALIZATION OF VFMS TO EM IMAGE DATA

To probe the extent to which general-purpose VFMs transfer to EM, we first benchmarked our 3 backbones under a frozen-backbone, segmentation head-only adaptation regime on Lucchi++ and VNC as described in 3.3.1. In Table 1, we report the $\text{IoU}_{\text{fg}}$ for the three training settings outlined in 3.3.3 (Lucchi++ only, VNC only, and paired training).

Across single-dataset training, all models achieve a relatively strong $\text{IoU}_{\text{fg}}$ on both Lucchi++ and VNC considering the nature of the frozen-backbone approach, indicating that pre-trained representations can support effective adaptation to a single EM dataset. In contrast, performance drops drastically under paired training for all backbones, despite the fact that both datasets share the same imaging modality. This degradation suggests that, while VFMs can yield useful representations for EM datasets taken in isolation, their representations do not readily support the training of a single unified model on heterogeneous EM image sources. To test whether the paired scenario drop is simply due to a strong dataset imbalance between Lucchi++ and VNC, we reran paired training with balanced sampling (Supplementary Section A.7). Our results indicate that balanced sampling does not improve performance, suggesting that the observed failure of paired training is due to inter-dataset domain mismatch and cannot be explained by dataset imbalance alone.

**Domain mismatch within the same modality.** To understand why training on paired data has such a major impact on performance despite both datasets being EM images, we analyzed the backbone feature space for Lucchi++ and VNC with DINOv2-L/14. In Figure 1a, we show a PCA projection of the image-level embeddings extracted from the frozen backbone. The two datasets occupy disjoint regions of the representation space, indicating that the pre-trained features encode strong dataset-specific signatures rather than a unified EM images manifold. This qualitative separation is corroborated quantitatively by a large Fréchet distance (79.32) between Gaussian approximations of the two feature distributions (FD-DINOv2), and by a near-perfect linear separability of the dataset label using an LR probe (accuracy and ROC-AUC of 1.0). Further analysis of the unlabelled stack (stack 2) of VNC yields similar results, indicating that these observations cannot be attributed to small-sample effects (Supplementary Section A.8). Together, these diagnostics indicate substantial domain mismatch between Lucchi++ and VNC at the representation level, which is consistent with the strong degradation observed when training a single decoder on the union of the two datasets.

### 4.2 IMPROVEMENT OF GENERALIZATION ACROSS EM DATASETS THROUGH PEFT

We next tested whether PEFT can mitigate the poor performance of VFMs observed under paired training. Using the same experimental protocol as in the previous section, we enable LoRA adapters while keeping the segmentation head unchanged. In Table 2, we report the $\text{IoU}_{\text{fg}}$ for single-dataset and paired training.

Across single-dataset training, LoRA consistently improves performance for all backbones by $\sim 10$ points relative to the frozen setting, indicating that limited backbone adaptation is beneficial when the training and test data come from a single EM source. However, paired training remains challenging: although LoRA yields a small improvement in the macro $\text{IoU}_{\text{fg}}$, the absolute performance remains poor for all backbones. These results suggest that PEFT can sharpen in-domain representations, but does not by itself enable a single natural image VFM to learn a robust shared representation that transfers across heterogeneous EM datasets presenting domain misalignment.

| Backbone | Lucchi++ | VNC | Paired |
|---|---|---|---|
| DINOv2-L/14 | $0.802 \pm 0.007$ | $0.818 \pm 0.003$ | $0.058 \pm 0.002$ |
| DINOv3-L/16 | $\mathbf{0.838 \pm 0.002}$ | $\mathbf{0.875 \pm 0.001}$ | $\mathbf{0.065 \pm 0.002}$ |
| OpenCLIP ViT-L/14 | $0.624 \pm 0.007$ | $0.679 \pm 0.017$ | $0.045 \pm 0.006$ |

Table 2: **LoRA-PEFT segmentation performance.** Foreground IoU ($IoU_{fg}$) on Lucchi++ and VNC under single-dataset and paired training with LoRA adapters in the backbone, reported as the mean±std over 5 runs. In the paired setting, the reported $IoU_{fg}$ is a macro-average across both datasets.

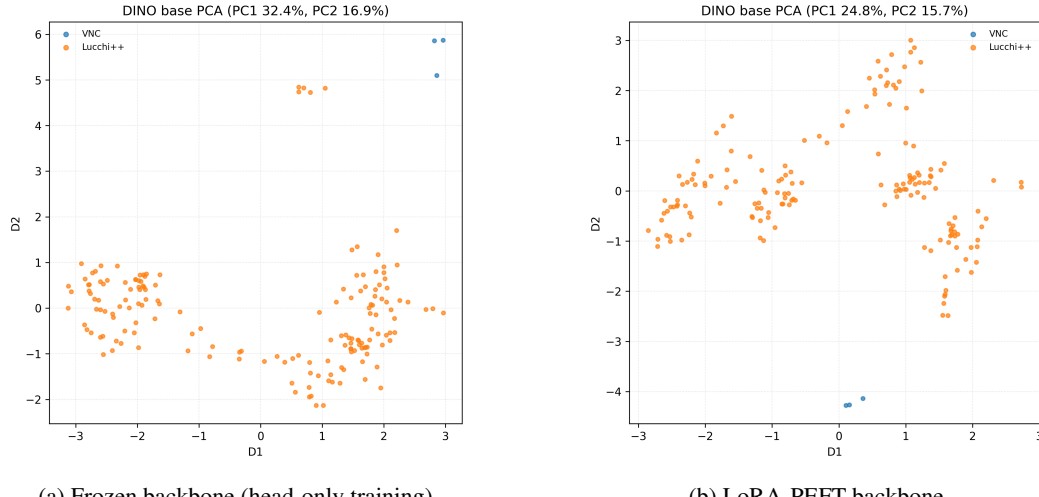

(a) Frozen backbone (head-only training)  (b) LoRA-PEFT backbone

Figure 1: **Domain mismatch between Lucchi++ and VNC in DINOv2 feature space.** (a) PCA projection of image-level embeddings from the frozen DINOv2-L/14 backbone exhibits no overlap between the two datasets. (b) With LoRA enabled, the feature-space gap is minimally reduced, and the two domains remain clearly separable. We also report the FD-DINOv2 (1) and LR performance for predicting the dataset label from the embedding vectors (frozen: $79.32$; LoRA: $45.02$; probe accuracy/AUROC $= 1.0$ in both).

**Domain mismatch is reduced but persists after PEFT.** To assess whether LoRA indeed aligns the two datasets in representation space, we repeated the domain mismatch diagnostics from 4.1 using embeddings extracted from the LoRA-adapted backbone. In Figure 1b, we observe that Lucchi++ and VNC are brought slightly closer from one another in the PCA projection as compared to the frozen backbone scenario, but remain clearly separable. Quantitatively, although the Fréchet distance computed in the DINOv2 feature space decreases from $79.32$ (frozen) to $45.02$ (LoRA), an LR probe is still able to predict the dataset label near-perfectly (accuracy/AUROC $= 1.0$). Further analysis of the unlabelled stack of VNC yields similar results, indicating that these observations cannot be attributed to small-sample effects (Supplementary Section A.8). Together, these results indicate that LoRA reduces but does not eliminate the representation gap between Lucchi++ and VNC, which is consistent with the minimal gains in performance observed in paired training.

### 4.3 PERFORMANCE OF VFMS WITH AND WITHOUT PEFT AGAINST STATE-OF-THE-ART ALTERNATIVES

Finally, we compare the quality of the VFM-based segmentation against representative state-of-the-art alternatives proposed in previous works on mitochondria segmentation on EM. In Figure 2, we report the $IoU_{fg}$ for supervised end-to-end baselines (*e.g.*, U-Net variants), prior prompt-based and zero-shot approaches (*e.g.*, CLIPSeg, ConvPaint), and for our VFM-based models with and without LoRA. To facilitate comparison with results reported in the literature, we have only evaluated our work on the Lucchi++ dataset and reserved the VNC dataset for the paired experiments.

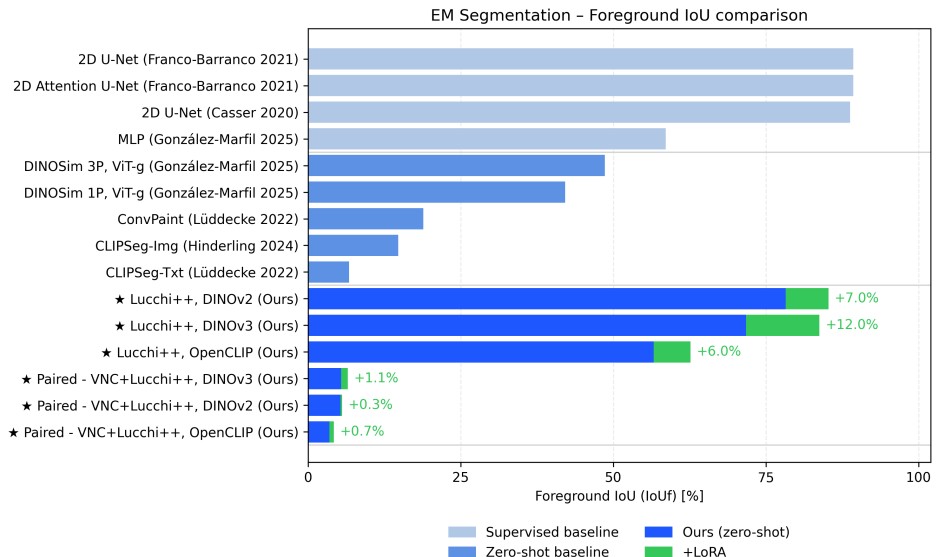

Figure 2: **Comparison to prior EM mitochondria segmentation approaches.** Foreground IoU ($IoU_{fg}$) for supervised baselines (top, light blue), zero-shot and prompt-based methods (middle, blue), and our VFM-based models with frozen (bottom, dark blue) and LoRA-adapted backbones (bottom, green). While VFMs are competitive in a single-dataset training scenario, performance collapses in the paired training setting, highlighting the impact of intra-EM domain mismatch.

As expected, fully supervised end-to-end baselines remain the strongest performers as they are optimized directly for their target dataset. In contrast, VFM-based models, even when enhanced with LoRA, do not manage to consistently match the best U-Net variants. Importantly, however, this performance gap comes with different computational and practical requirements: while end-to-end training requires substantially more task-specific optimization and compute, frozen-backbone adaptation and PEFT offer a general way to only update a small fraction of parameters while reusing the pre-trained representation. From this perspective, VFMs can offer an attractive option when compute or annotation budgets are constrained, or when quick adaptation to a new dataset is desired.

The paired setting, however, highlights a more fundamental limitation of VFMs. While these models can transfer reasonably well to individual microscopy image datasets despite being pre-trained predominantly on natural images, they fail to yield a single model that performs well on the union of different datasets from EM imaging. In our experiments, we observe that performance under paired training is dramatically worse than in any single-dataset setting, and that LoRA only offers a marginal improvement. This suggests that, besides the mismatch between natural images and EM data, a substantial *intra-modality* shift within EM may be the main hindrance to generalization. When two datasets of the same microscopy imaging modality are sufficiently far apart in representation space, a lightweight adaptation therefore appears to be insufficient to align them into a shared feature space and to support robust segmentation performance across both of them.

## 5 CONCLUSION

Across three widely used VFMs (DINOv2, DINOv3 and OpenCLIP), we found that pre-trained representations can be effectively reused for mitochondria segmentation on EM image data when training and testing within a single dataset. While a simple decoder added on a frozen backbone already yields strong performance, a refinement of the backbone with LoRA further improves in-domain performance. However, when attempting to train a single "general EM" model on the union of two EM datasets, we observe a collapse of performance for all considered backbones, with PEFT only providing marginal improvement. Investigations of the latent representations of the models considered, relying on PCA projections, calculation of the Fréchet distance in DINOv2 feature space, and linear probing, indicate that the EM datasets remain strongly separable even after PEFT, suggesting

that intra-modality domain shift in microscopy image datasets is a major challenge. While dataset-specific intensity normalization might be a promising strategy to reduce the observed domain shift, prior work on EM segmentation has shown that standard approaches such as contrast-limited adaptive histogram equalization (CLAHE) fail to resolve performance collapse across datasets (Gallusser et al., 2022), suggesting that differences extend beyond first-order intensity statistics. Overall, our results hint at the fact that current VFMs are not yet foundational enough to handle the variability inherent in microscopy imaging, even when only considering EM imaging data only. They also reveal that existing mechanisms for domain alignment, primarily developed for natural images, may not suffice in other, more variable image domains such as microscopy. These observations call for further methodological developments to support the efficient adaptation of VFMs to microscopy data and to ultimately enable the development of models that generalize over imaging modalities.

## MEANINGFULNESS STATEMENT

A key challenge in learning meaningful representations of life is to ensure that representations encode biological insight rather than dataset-specific signatures. We consider 3 widely-used vision foundation models and investigate, through a segmentation task, whether they can encode the semantics of electron microscopy images in a way that generalizes across distinct datasets. Our results suggest that current models trained on natural images provide meaningful representations on isolated microscopy datasets, but are not yet robust enough when multiple datasets from the same imaging modality are combined. This highlights a limitation for biological representation learning and motivates future work on domain alignment.

## ACKNOWLEDGMENTS

The authors were supported by the University of Zurich.

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

# A SUPPLEMENTARY MATERIAL

## A.1 DATASET OVERVIEW

**VNC (Drosophila ssTEM) (Gerhard et al., 2013).** The VNC dataset is commonly used to study mitochondria segmentation. It consists of two sequential serial-section transmission electron microscopy (ssTEM) image stacks of the *Drosophila melanogaster* third-instar larva ventral nerve cord (VNC), each composed of $1024 \times 1024 \times 20$ voxels. The first stack provides multiple object labels, including binary mitochondria segmentation masks (foreground mitochondria vs. background), while the second one contains raw images only. Following prior work using this dataset (González-Marfil et al., 2025), we used the annotated stack as our supervised benchmark and split its 20 slices into 17 training and 3 test images. The second (unlabelled) stack was reserved for unsupervised analyses, as it does not include any ground-truth annotations.

**Lucchi++ (Mouse Hippocampus FIB-SEM) (Lucchi et al., 2011; Casser et al., 2020).** The Lucchi++ dataset is a focused ion beam scanning electron microscopy (FIB-SEM) image volume of the mouse hippocampus (CA1) acquired at an isotropic 5nm resolution, and is commonly used for mitochondria segmentation benchmarks. In contrast to VNC, the Lucchi++ dataset comes with a predefined train/test split and corresponding annotations. Specifically, the dataset includes two sequential stacks for training and testing, with 165 images per split (*i.e.*, 165 annotated training images and 165 annotated test images), yielding four sequential stacks in total. Each slice is composed of $1024 \times 768$ pixels and is accompanied by a binary mitochondria segmentation mask (foreground mitochondria vs. background), which we use as sole supervision signal in our experiments. We adopt this standard split throughout our experiments with this dataset.

In Figure 3, we illustrate representative images of the two datasets.

## A.2 RATIONALE FOR THE CHOICE OF VISION FOUNDATION MODEL BACKBONES

We here motivate the choice of backbones for transfer to our microscopy image data analysis application of focus.

First, we selected DINOv2 (Oquab et al., 2023), which appeared as a natural starting point as recent work reported promising performance on EM segmentation leveraging DINOv2 features, albeit under a different adaptation strategy than ours (González-Marfil et al., 2025).

Second, considering the recent release of DINOv3 (Siméoni et al., 2025) and its emerging adoption in microscopy image analysis settings through parameter-efficient adaptation on other modalities (Muminov & Pham, 2025; Balezo et al., 2025), we tested whether this newer model improves the robustness and transfer of the data representation as compared to DINOv2 under the same controlled protocol. Finally, we also included OpenCLIP (Cherti et al., 2023) to probe a qualitatively different pre-training signal.

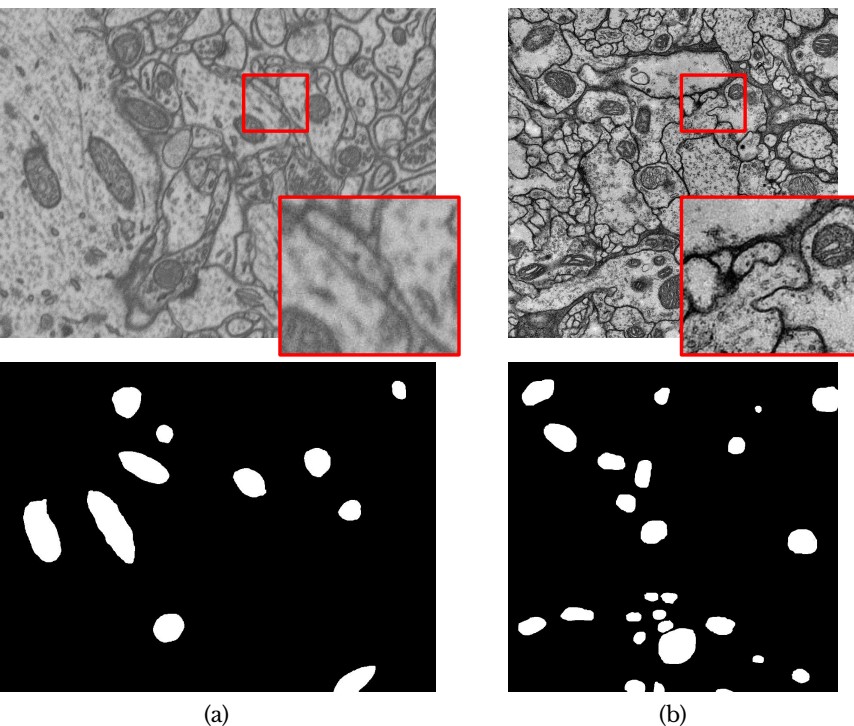

(a)                                                    (b)

Figure 3: **Illustration of the two EM datasets used in this work.** Representative slices from (a) Lucchi++ (mouse hippocampus, FIB-SEM) and (b) VNC (Drosophila ventral nerve cord, ssTEM), shown with their corresponding binary mitochondria ground-truth masks. The red box and inset highlight a local region to emphasize differences in texture/contrast and membrane appearance across datasets, despite both being EM images.

OpenCLIP has not been broadly studied for EM segmentation, and only limited benchmarking evidence in microscopy data is available with this model to date (Lozano et al., 2024). It is a vision-language architecture, although we do not consider the language part in this work. As such, it offers an opportunity to evaluate whether a backbone with such a different architecture provides any advantage (or failure mode) under microscopy data domain shift.

All three backbones are pre-trained on a large corpus of natural images (*i.e.*, not microscopy data), but with different objectives and data sources: DINOv2 is trained in a self-supervised manner on the curated LVD-142M web dataset, DINOv3 is trained on a curated subset derived from a multi-billion pool of public web images, and the OpenCLIP variant we use is trained contrastively on the LAION-2B subset of LAION-5B. Our working hypothesis is that the scale and diversity of these pre-training paradigms yield broadly reusable visual primitives and robust representations that can transfer to microscopy images with limited supervision. Our experiments aim to explicitly test this assumption and to assess the extent to which parameter-efficient adaptation facilitates the transfer to EM imaging data.

All backbones are ViT-based and thus operate on a fixed patch size, which determines the token grid and constrains input handling. DINOv2 is released as ViT-·/14, and we therefore standardize our preprocessing to be compatible with a $14 \times 14$ patchification, as described in Section 3.2.2. DINOv3 uses ViT-·/16 and its implementation is more permissive to varying input sizes (*e.g.*, through positional-embedding handling), reducing the need for strict input divisibility. OpenCLIP is available in both /14 and /16 variants. Since our primary OpenCLIP backbone is ViT-L/14 and to avoid confounding tokenization geometry with model choice, we consistently use the /14 OpenCLIP variants.

## A.3 Effect of backbone scale on segmentation performance

In Tables 3, 4, and 5, we report the $IoU_{fg}$ for multiple backbone sizes (ViT-S: small, ViT-B: big, ViT-L: large, and ViT-G: giant) under the two adaptation regimes considered (frozen-backbone training and LoRA-PEFT). Across all three analyzed VFMs, scaling the backbone generally yields incremental gains in the single-dataset setting, and LoRA consistently improves performance compared to the frozen baseline for a fixed backbone size. However, our main conclusion remains unchanged across scales: training on the union of the Lucchi++ and VNC datasets leads to a severe degradation in $IoU_{fg}$, and increasing the backbone size does not rescue performance.

| DINOv2 | Lucchi++ ($IoU_{fg}$) | | VNC ($IoU_{fg}$) | | Paired ($IoU_{fg}$) | |
|---|---|---|---|---|---|---|
| | **Frozen** | **LoRA** | **Frozen** | **LoRA** | **Frozen** | **LoRA** |
| ViT-S/14 | $0.606 \pm 0.008$ | $0.731 \pm 0.007$ | $0.769 \pm 0.004$ | $0.825 \pm 0.001$ | $\mathbf{0.058 \pm 0.006}$ | $0.058 \pm 0.004$ |
| ViT-B/14 | $0.716 \pm 0.009$ | $0.774 \pm 0.005$ | $0.757 \pm 0.006$ | $0.857 \pm 0.004$ | $0.055 \pm 0.003$ | $0.049 \pm 0.001$ |
| ViT-L/14 | $0.717 \pm 0.006$ | $0.802 \pm 0.007$ | $0.723 \pm 0.002$ | $0.818 \pm 0.002$ | $0.052 \pm 0.005$ | $\mathbf{0.058 \pm 0.002}$ |
| ViT-G/14 | $\mathbf{0.783 \pm 0.002}$ | $\mathbf{0.852 \pm 0.003}$ | $\mathbf{0.810 \pm 0.003}$ | $\mathbf{0.860 \pm 0.003}$ | $0.053 \pm 0.001$ | $0.056 \pm 0.003$ |

Table 3: **Segmentation performance of DINOv2 variants.** Foreground IoU ($IoU_{fg}$) in the frozen-backbone (head-only) and LoRA-PEFT setting reported as the mean±std over repeated runs.

| DINOv3 | Lucchi++ ($IoU_{fg}$) | | VNC ($IoU_{fg}$) | | Paired ($IoU_{fg}$) | |
|---|---|---|---|---|---|---|
| | **Frozen** | **LoRA** | **Frozen** | **LoRA** | **Frozen** | **LoRA** |
| ViT-S/16 | $0.598 \pm 0.000$ | $0.722 \pm 0.009$ | $0.772 \pm 0.000$ | $0.831 \pm 0.002$ | $0.051 \pm 0.000$ | $\mathbf{0.070 \pm 0.003}$ |
| ViT-B/16 | $0.646 \pm 0.000$ | $0.775 \pm 0.004$ | $0.772 \pm 0.000$ | $0.828 \pm 0.002$ | $0.044 \pm 0.000$ | $0.060 \pm 0.001$ |
| ViT-L/16 | $\mathbf{0.717 \pm 0.000}$ | $\mathbf{0.838 \pm 0.002}$ | $\mathbf{0.833 \pm 0.000}$ | $\mathbf{0.875 \pm 0.001}$ | $\mathbf{0.054 \pm 0.000}$ | $0.065 \pm 0.002$ |

Table 4: **Segmentation performance of DINOv3 variants.** Foreground IoU ($IoU_{fg}$) in the frozen-backbone (head-only) and LoRA-PEFT setting reported as the mean±std over 5 repeated runs.

## A.4 LoRA-PEFT configuration and trainable parameters

To make our PEFT setting fully explicit, we here detail the specific backbone parameters we adapt and how. Across all backbones and experimental regimes, the segmentation head architecture and its training procedure are kept fixed. The only additional trainable parameters introduced under PEFT

| OpenCLIP | Lucchi++ (IoU$_{fg}$) | | VNC (IoU$_{fg}$) | | Paired (IoU$_{fg}$) | |
|---|---|---|---|---|---|---|
| | Frozen | LoRA | Frozen | LoRA | Frozen | LoRA |
| ViT-L/14 | **0.568 ± 0.003** | **0.624 ± 0.007** | 0.573 ± 0.011 | 0.679 ± 0.017 | 0.032 ± 0.003 | 0.045 ± 0.006 |
| ViT-H/14 | 0.556 ± 0.003 | 0.610 ± 0.011 | **0.578 ± 0.008** | **0.687 ± 0.012** | **0.039 ± 0.006** | **0.048 ± 0.003** |

Table 5: **Segmentation performance of OpenCLIP variants.** Foreground IoU (IoU$_{fg}$) in the frozen-backbone (head-only) and LoRA-PEFT setting reported as the mean±std over repeated runs.

| Backbone | Patch size | LoRA targets | $r$ | $\alpha$ | Trainable parameters | Frozen parameters |
|---|---|---|---|---|---|---|
| DINOv2 | 14 | attn qkv + attn proj | 16 | 32 | head + LoRA $\mathbf{A}, \mathbf{B}$ | backbone |
| DINOv3 | 16 | attn qkv + attn proj | 16 | 32 | head + LoRA $\mathbf{A}, \mathbf{B}$ | backbone |
| OpenCLIP | 14 | attn qkv + attn proj | 16 | 32 | head + LoRA $\mathbf{A}, \mathbf{B}$ | backbone |

Table 6: **PEFT configurations by backbone.** We apply LoRA to attention projections only (qkv and output projection). Under PEFT, we only optimize the segmentation head and LoRA adapter matrices, while all the original backbone weights remain frozen.

are the LoRA adapter weights injected inside the backbone. Concretely, we apply LoRA to the ViT attention projections only (query/key/value and output projection), while keeping all original backbone weights frozen. We provide a summary of the details of the PEFT configuration for each backbone in Table 6.

**Trainable vs. frozen parameters under PEFT.** Under PEFT, the set of trainable parameters consists of the segmentation head parameters and the low-rank LoRA matrices $\mathbf{A}$ and $\mathbf{B}$ attached to each targeted linear projection. All original backbone weights remain frozen after LoRA injection. Formally, for a targeted linear map with weight $\mathbf{W}$, LoRA replaces it with $\mathbf{W}' = \mathbf{W} + \frac{\alpha}{r}\mathbf{BA}$, where $r$ is the LoRA rank, $\alpha$ is the scaling factor, and $\mathbf{A}, \mathbf{B}$ are the two LoRA matrices being optimized.

**LoRA hyperparameters and patch size.** In all runs, we used $r = 16$ based on small preliminary sweeps and following common practice in the PEFT literature. Increasing $r$ is typically expected to improve performance but also increases the number of trainable parameters linearly, bringing the method closer to full fine-tuning and reducing the parameter-efficiency relevance (Hu et al., 2022; Dettmers et al., 2023). Similarly, we set the scaling $\alpha = 32$ following standard LoRA conventions to stabilize the effective update magnitude across ranks (Agiza et al., 2024; Hugging Face, 2026).

The backbone patch sizes are fixed by the pre-trained model variants: DINOv2 uses patch size 14, DINOv3 uses patch size 16, and the OpenCLIP variants used in this work use patch size 14.

**Parameter count reporting.** For transparency, each run writes a `lora_targets.json` report containing the resolved LoRA targets and the corresponding trainable parameter counts (total trainable, LoRA-only, and head-only). We used these reports to verify that only the intended modules were optimized.

### A.5 TRAINABLE PARAMETER BUDGET UNDER LIGHTWEIGHT ADAPTATION

To contextualize the accuracy-efficiency trade-off of lightweight adaptation, we report in Table 7 the number of trainable parameters for head-only training versus LoRA-PEFT (LoRA adapters plus segmentation head) for several backbones. As expected, head-only adaptation keeps the trainable parameter budget nearly constant across backbone sizes, since only the decoder head is optimized. In contrast, enabling LoRA increases the trainable parameter count with model scale. These numbers highlight that PEFT updates only a small fraction of the backbone while still yielding consistent in-domain gains, as seen in Tables 3 and 4.

Notably, the reported trainable-parameter counts are identical for the matched DINOv2 and DINOv3 variants under our configuration. This is expected as the trainable budget in our setup is determined solely by the adaptation modules, namely the segmentation head and, when enabled, the LoRA adapters, whose parameterization depends on the backbone feature dimensionality (and on the set of adapted linear layers) rather than on the input image size. Since we apply the same LoRA configu-

| Backbone | Variant | Trainable parameters (head-only) | Trainable parameters (LoRA+head) |
|----------|---------|----------------------------------|----------------------------------|
| DINOv2 | ViT-S/14 | 3.829M (14.8%) | 4.271M (16.5%) |
| DINOv2 | ViT-B/14 | 4.025M (4.4%) | 4.910M (5.4%) |
| DINOv2 | ViT-L/14 | 4.157M (1.3%) | 6.516M (2.1%) |
| DINOv2 | ViT-G/14 | 4.419M (0.4%) | 10.317M (0.9%) |
| DINOv3 | ViT-S/16 | 3.829M (15.1%) | 4.271M (16.8%) |
| DINOv3 | ViT-B/16 | 4.025M (4.5%) | 4.910M (5.5%) |
| DINOv3 | ViT-L/16 | 4.157M (1.4%) | 6.516M (2.1%) |
| OpenCLIP | ViT-L/14 | 4.157M (1.3%) | 6.516M (2.1%) |
| OpenCLIP | ViT-H/14 | 4.288M (0.7%) | 8.220M (1.3%) |

Table 7: **Trainable parameter counts under various lightweight adaptation strategies.** We report the number of trainable parameters when training only the segmentation head (frozen backbone, head-only) versus enabling LoRA-PEFT in the backbone (LoRA+head, rank $r = 16$, $\alpha = 32$, applied to attention `qkv` and `proj`). Percentages in parentheses indicate the fraction of trainable parameters relative to the total number of parameters in the corresponding end-to-end model. The number of trainable parameters does not depend on the dataset considered for a fixed backbone and output label configuration.

ration (rank $r$=16, $\alpha$=32, targets `attn.qkv` and `attn.proj`) and the same head architecture to the DINOv2 and DINOv3 variants with matching embedding dimensions, the resulting number of trainable parameters coincides.

## A.6 EVALUATION METRICS

**Fréchet DINOv2 Distance** To quantitatively assess the mismatch between different data distributions, we rely on the Fréchet DINOv2 Distance (FD-DINOv2, Stein et al. (2023)), calculated as follows. Given two sets of pooled embeddings $E_1$ and $E_2$, we approximate each $E_i$ by a multivariate Gaussian $\mathcal{N}(\mu_i, \Sigma_i)$ with empirical mean $\mu_i$ and covariance $\Sigma_i$, and compute

$$\text{FD-DINOv2}(E_1, E_2) = \|\mu_1 - \mu_2\|_2^2 + \text{Tr}\left(\Sigma_1 + \Sigma_2 - 2(\Sigma_1^{1/2}\Sigma_2\Sigma_1^{1/2})^{1/2}\right). \tag{1}$$

**Foreground IoU** To quantitatively evaluate segmentation performance, we use the foreground IoU, calculated as follows. With $p$ and $m$ denoting the predicted and ground-truth label maps, respectively, and considering the foreground indicators $p_{\text{fg}} = \mathbb{1}[p > 0]$ and $m_{\text{fg}} = \mathbb{1}[m > 0]$, the foreground IoU is defined as

$$\text{IoU}_{\text{fg}} = \frac{|p_{\text{fg}} \cap m_{\text{fg}}|}{|p_{\text{fg}} \cup m_{\text{fg}}| + \epsilon}, \tag{2}$$

where $\epsilon$ is a small constant included for numerical stability ($1e - 7$ in our experiments). Ground-truth masks can optionally be binarized at load time using a fixed threshold of 128, corresponding to the midpoint for 8-bit $[0, 255]$ masks (*i.e.*, values $\geq 128$ are mapped to foreground and $< 128$ to background), which affects $m$ (and therefore $m_{\text{fg}}$) in the corresponding experiments. Model predictions are obtained by taking an `argmax` over the per-pixel logits.

## A.7 EFFECT OF THE UNBALANCED NATURE OF THE VNC DATASET ON PAIRED TRAINING

The paired training setting is inherently imbalanced as VNC provides substantially fewer labelled images than Lucchi++. To test whether the severe performance collapse observed under paired training is driven by this imbalance, we run an additional ablation with the frozen-backbone, head-only DINOv2-L/14 configuration. We compare our default unbalanced approach (standard concatenation and shuffling) against a balanced strategy that enforces $1 + 1$ batching. This ensures that each mini-batch contains one Lucchi++ sample and one VNC sample, effectively upsampling the smaller dataset. In Table 8, we report $\text{IoU}_{\text{fg}}$ separately on each test set as well as their macro-average.

Our balancing strategy modestly increases the macro $\text{IoU}_{\text{fg}}$, but the absolute performance remains poor. The model remains effectively unable to segment Lucchi++ in the paired setting, as seen by

| Sampling | Lucchi++ $\text{IoU}_{\text{fg}}$ | VNC $\text{IoU}_{\text{fg}}$ | Paired $\text{IoU}_{\text{fg}}$ |
|---|---|---|---|
| Unbalanced | $0.007 \pm 0.004$ | $0.661 \pm 0.022$ | $0.050 \pm 0.002$ |
| Balanced | $\mathbf{0.019 \pm 0.003}$ | $\mathbf{0.732 \pm 0.015}$ | $\mathbf{0.064 \pm 0.003}$ |

Table 8: **Paired-training failure is not resolved by dataset balancing.** Foreground IoU ($\text{IoU}_{\text{fg}}$) segmentation performance of a frozen-backbone DINOv2-L/14 with head-only training in the paired dataset (Lucchi++ $\cup$ VNC) training regime reported as the mean±std over repeated runs. The balanced setting implements $1 + 1$ batching to ensure that each mini-batch contains one sample per dataset.

the Lucchi++ $\text{IoU}_{fg}$ being close to $0$. Interestingly, the model achieves a comparatively high $\text{IoU}_{\text{fg}}$ on VNC despite its under-representation, indicating that the paired-training performance collapse cannot be explained by dataset imbalance alone. This is consistent with our representation-space diagnostics, which suggest that a strong inter-dataset domain mismatch is the dominant factor limiting joint training.

### A.8 DOMAIN MISMATCH DIAGNOSTICS ON UNLABELLED VNC STACK2

In Sections 4.1 and 4.2, we quantify representation-space domain mismatch between Lucchi++ and VNC using the labelled VNC split (Stack1), following the same supervised setting used throughout the segmentation experiments. However, this labelled test subset contains only 3 slices, which makes distribution-level diagnostics such as feature-space distances potentially sensitive to small-sample effects. Since our domain-mismatch analysis is fully unsupervised and does not require masks, we repeated the experiment on the unlabelled VNC Stack2, which provides a larger set of 20 images for evaluation.

Concretely, we extracted one image-level embedding per slice from DINOv2 by global pooling of the final-layer patch tokens, as in Figure 1, and then carried out PCA on the joint set of Lucchi++ test images and VNC Stack2 slices. In Figure 4, we visualize the first two principal components in the frozen backbone (head-only training) and after applying LoRA-PEFT. To complement this visualization, we also report the FD-DINOv2 (equation 1) and the linear separability of the domain label (Lucchi++ vs. VNC) using an LR probe.

We observe that data points from Lucchi++ and the VNC Stack2 remain clearly separated in feature space, which is consistent with the conclusions reached considering the labelled VNC split. Quantitatively, the FD-DINOv2 remains large for the frozen backbone and decreases when LoRA is applied ($82.06$ and $43.26$, respectively), but the two datasets are still perfectly linearly separable (accuracy/AUROC = $1.0$ in both cases). Overall, increasing the number of VNC images from a handful of labelled test slices to the full unlabelled Stack2 therefore does not alter the qualitative outcome: the domain mismatch between the two datasets persists and is only marginally reduced by LoRA, supporting the interpretation that large inter-dataset differences are the main barrier for paired training.

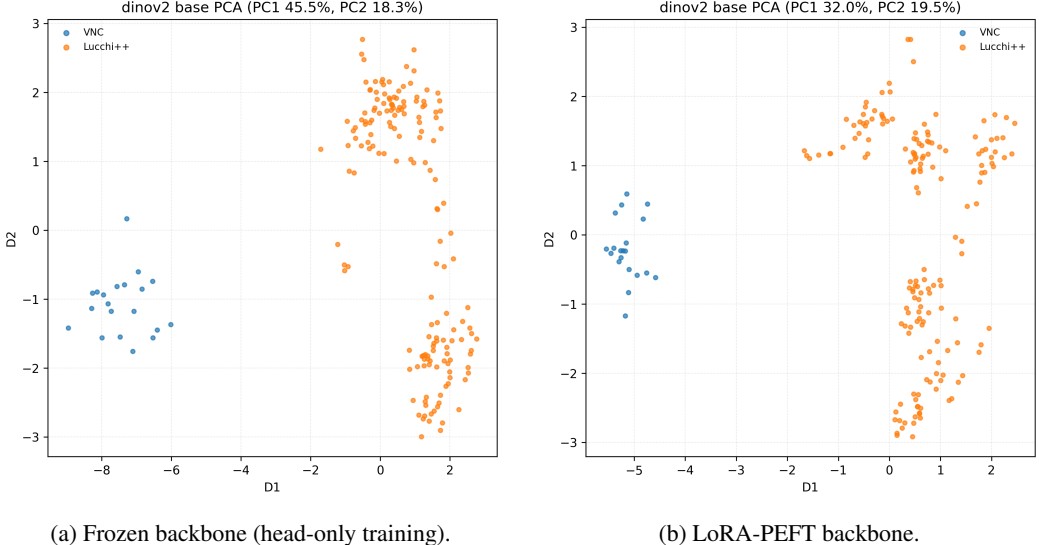

(a) Frozen backbone (head-only training).      (b) LoRA-PEFT backbone.

Figure 4: **Domain mismatch between Lucchi++ and VNC (Stack2) in DINOv2 feature space.** (a) PCA projection of image-level embeddings from the frozen DINOv2-L/14 backbone exhibits no overlap between the two datasets. (b) With LoRA enabled, the feature-space gap is minimally reduced, and the two domains remain clearly separable.

