# OpenReview forum: "Are Vision Foundation Models foundational for electron microscopy image segmentation?"
_ICLR.cc/2026/Workshop/LMRL — ICLR 2026 Workshop LMRL Poster_

### Official Review · Reviewer_WvNU · 2026-02-25

**Rating:** 6
**Confidence:** 3

**Review:**

### Summary

The authors evaluate if foundational vision models can provide latent representations that can be transferred across EM datasets. They test their adaptation framework on two settings, one with a frozen backbone training with a segmentation head and the other using PEFT based finetuning via LoRA while keeping the segmentation head frozen. They show that  best performance in terms IoU is observed when trained on a single EM dataset but pooling multiple EM sets causes severe decrease in performance. They investigate further on the performance dip and conclude the poor joint training  performance as seen while analyzing pooled backbone embeddings through PCA and frechet distance.

### Strengths:
- The authors  provides a clear motivation addressing the current challenged in learning invariant microscopy representations framing the problem to be distribution shift in the same modality.

- The authors have done ample experiments and benchmarked the framework against current methods and conclude that fully supervised methods remain as the best choice when tuned for a particular datasetss.


### Weaknesses/Cons:
- There seems to be inconsistency in the results reported in Table 1 (around 0.053 for DINOv2)  and Table 8 (much higher values) for the paired training regime. Seems like computation for paired and averaging seems to be different in the evaluation but are not explained by the authors.

- The paired training setup uses resizing strategies to handle multiple input dimensions but this can degrade performance due to potential noise or information loss that can occur in the preprocessing stage.

- Validation and testing samples are very low, showing standard deviation doesn't mean much when evaluated on such a small subset. This is important as the std reported in most the fields were very low.

---

### Official Review · Reviewer_7QLS · 2026-02-25
**Domain Mismatch Limits Paired-Dataset Fine-Tuning of Vision Foundation Models for Mitochondria Segmentation**

**Rating:** 7
**Confidence:** 3

**Review:**

## Summary

This manuscript studies whether standard fine-tuning strategies for vision foundation models (VFMs), specifically Low-Rank Adaptation (LoRA) and lightweight segmentation-head training, can improve performance on mitochondria segmentation in electron microscopy (EM). The authors benchmark three VFMs (DINOv2, DINOv3, and OpenCLIP) on two EM datasets under (i) a frozen-backbone setup with a small decoder head and (ii) parameter-efficient fine-tuning via LoRA.

Across experiments trained on a single dataset, the adapted VFMs achieve reasonably strong foreground IoU and, in general, LoRA improves in-domain performance. However, the VFM-based approaches remain slightly below dedicated state-of-the-art task-specific methods when evaluated on a single dataset.

A particularly interesting result is that performance collapses when models are trained on the combined (paired) datasets. The paper backs this observation with a thorough representation-space analysis suggesting a persistent domain mismatch between datasets, which plausibly explains the failure of paired training. Overall, the empirical analysis is convincing, and the domain-mismatch finding is both interesting and potentially impactful for future method design.

## Comments and questions for future work

- It would be informative to evaluate paired-training performance as a function of the amount of data from each dataset. This could clarify whether the issue can be mitigated by scaling data in the target domain.

- Replicate figure 2 on the second dataset.

- Is the degraded paired performance primarily caused by the fine-tuning objective/adaptation strategy, or by limitations of the VFM architectures themselves?
     - If computationally feasible, fully fine-tuning the backbone (all parameters) could help isolate whether PEFT constraints are a key limiting factor. Alternatively, training the same architecture from scratch (random initialization) on the paired datasets could help distinguish pre-training-induced biases from architectural constraints.
     - If full-size training is too expensive, a practical compromise would be to train smaller/lightweight architectures (e.g., representative baselines similar to those compared in the paper; figure 2) from scratch on the paired datasets and test whether they also exhibit the same collapse. This would help assess whether the paired-training failure is specific to VFMs/transfer or a broader objective/domain issue.

---

### Official Review · Reviewer_2cco · 2026-02-25

**Rating:** 6
**Confidence:** 4

**Review:**

This paper investigates whether vision foundation models (DINOv2, DINOv3, OpenCLIP) learn representations that are general enough to transfer across different electron microscopy datasets for mitochondria segmentation. The results show that while both frozen-backbone training and LoRA-based fine-tuning perform well within a single dataset, performance drops significantly when training jointly on multiple datasets, suggesting that current PEFT methods are not sufficient to overcome strong cross-dataset domain shifts without additional alignment strategies.

Advantages:
- The paper provides a thorough architectural comparison, evaluating several modern VFMs under both frozen and parameter-efficient fine-tuning setups.
- It strengthens its claims by analyzing the latent representation space (PCA visualization, Fréchet DINOv2 distance, linear separability of domains), rather than relying solely on segmentation metrics.

Disadvantages:
- Although both datasets focus on mitochondria, they differ substantially in imaging modality (serial-section TEM vs. FIB-SEM), organism (fly vs. mouse), and likely hardware and acquisition settings. The observed domain mismatch may therefore stem from low-level acquisition artifacts rather than higher-level semantic differences, even if the images appear visually similar.
- The study only evaluates large ViT-based backbones. Including a smaller baseline model (e.g., ResNet-18 trained from scratch) would help clarify whether foundation models truly offer an advantage in this limited-data biomedical setting.

---

### Meta-Review · Area_Chair_kF9o · 2026-02-28

**Recommendation:** Accept (Poster)
**Confidence:** 4

**Metareview:**

Accept

---

### Decision · Program_Chairs · 2026-03-02

**Decision:**

Accept (Poster)

**Comment:**

Please see the meta-review.